# The Impact of Intestinal Microbiota and Toll-like Receptor 2 Signaling on α-Synuclein Pathology in Nontransgenic Mice Injected with α-Synuclein Preformed Fibrils

**DOI:** 10.3390/microorganisms12010106

**Published:** 2024-01-05

**Authors:** Yukako Koyanagi, Momoe Kassai, Hiroshi Yoneyama

**Affiliations:** 1Laboratory of Animal Microbiology, Department of Microbial Biotechnology, Graduate School of Agricultural Science, Tohoku University, Sendai 980-0845, Japan; yukako.koyanagi.q7@dc.tohoku.ac.jp; 2Sumitomo Pharma Co., Ltd., Osaka 554-0022, Japan

**Keywords:** intestinal microbiota, Toll-like receptor 2, Parkinson’s disease, animal model, α-synuclein

## Abstract

Intestinal microbiota and Toll-like receptor 2 (TLR2), which can bind lipoteichoic acid produced by microbiota, might contribute to the pathogenesis of Parkinson’s disease (PD), which is characterized by α-synuclein accumulation. Although the contribution of intestinal microbiota and TLR2 to PD pathology was validated in genetic PD models, evidence suggests that the effects of TLR2 signaling on proteinopathy might depend on the presence of a genetic etiology. We examined the impact of intestinal microbiota and TLR2 signaling on α-synuclein pathology in a nontransgenic mouse model of sporadic PD. While an α-synuclein preformed fibrils injection successfully reproduced PD pathology by inducing accumulation of α-synuclein aggregates, microglial activation and increased TLR2 expression in the brains of nontransgenic mice, antibiotic-induced reduction in the density of intestinal microbiota and TLR2 knockout had small impact on these changes. These findings, which are in contrast to those reported in transgenic mice harboring transgene encoding α-synuclein, indicate that the contribution of intestinal microbiota and TLR2 signaling to α-synuclein pathogenesis might be influenced by the presence of a genetic etiology. Additionally, these findings suggest that integrating insights from this experimental model and genetic models would further advance our understanding of the molecular mechanisms underlying sporadic PD.

## 1. Introduction

Parkinson’s disease (PD), the second most common neurodegenerative disease, is characterized by motor symptoms, including bradykinesia with tremor and/or muscle rigidity, and histologic changes, including the accumulation of Lewy bodies predominantly composed of α-synuclein and the pathologic loss of dopaminergic neurons in substantia nigra [1]. Approximately 26% of PD cases also have dementia, which affects their health-related quality of life and mortality [2,3,4]. Although the prevalence of dementia increases with the progression of the disease, there is no effective treatment for disease-modifying therapies that could delay or slow down the progression of the disease [2,5]. One of the contributing factors to the absence of disease-modifying therapy is the incomplete understanding of the molecular mechanisms underlying PD pathogenesis. The accumulation of aggregated α-synuclein is speculated to contribute to the pathogenesis of PD, which is therefore considered a proteinopathy [6]. Due to the prion-like behavior of pathological α-synuclein, they can propagate from cell to cell, leading to their accumulation in the widespread brain regions in the late stage of PD [7,8,9]. The severity of α-synuclein pathology in the cortical areas correlates with cognitive decline in PD [10]. Therefore, inhibition of α-synuclein accumulation could provide an approach to developing disease-modifying therapies for PD. However, the mechanisms underlying α-synuclein accumulation during disease progression are not fully understood. Specifically, nongenetic factors might contribute to the pathogenesis of sporadic PD, unlike familial PD, which is linked to genetic causes such as mutations in *SNCA* encoding α-synuclein [11].

Inflammation is a nongenetic factor that is considered to contribute to PD pathogenesis based on several observations. Autoimmune diseases and inflammatory bowel disease are risk factors for PD, and anti-inflammatory therapy for inflammatory bowel disease reduces the risk for PD [12,13,14]. Additionally, induction of intestinal inflammation using dextran sodium sulfate leads to the activation of microglia and α-synuclein accumulation in the brain [15,16,17]. Furthermore, levels of inflammatory factors in the brain and peripheral blood are altered in patients with PD [18,19,20]. Altogether, these findings suggest that alterations in inflammatory response might play a role in PD pathogenesis through interactions with α-synuclein.

Regarding inflammation in PD pathogenesis, intestinal microbiota and Toll-like receptor 2 (TLR2), a pattern recognition receptor that, once activated, induces the production of inflammatory cytokines [21], have been suggested to play functional roles in PD pathogenesis. This hypothesis is supported by several observations. First, intestinal dysbiosis observed in PD patients may contribute to increased intestinal and blood–brain barrier permeability in the patients [22]. Second, reducing the density of intestinal microbiota in the *SNCA* transgenic mouse model of PD suppresses the pathologic inflammatory responses and the accumulation of α-synuclein aggregates in the brain, whereas the transplantation of intestinal microbiota from patients with PD to animals worsens the PD pathology [23]. Third, the copy number of TLR2 ligand synthesis genes is increased in the intestinal microbiota of patients with PD [24]. Finally, a reduction in the density of intestinal microbiota leads to a reduction in TLR2 expression in the brain [25]. Considering that α-synuclein aggregates act as ligands for TLR2 [26], it is possible that alterations in intestinal microbiota and increased accumulation of α-synuclein aggregates might lead to TLR2 activation, contributing to PD pathogenesis. This hypothesis is supported by the increased expression of TLR2 observed in the brain and peripheral tissues of patients with PD [27,28,29] and the suppression of pathologic changes in inflammatory responses and α-synuclein aggregate accumulation in the brains of TLR2-knockout (TLR2-KO) in the *SNCA* transgenic mice [23].

These potential roles of intestinal microbiota and TLR2 signaling in PD pathogenesis are based on findings from *SNCA* transgenic mice, which model the genetic etiology of PD. *SNCA* transgenic mice successfully reproduce the pathologic changes found in patients with familial PD carrying *SNCA* mutations, as well as those observed in patients with sporadic PD. However, sporadic PD does not necessarily have a genetic etiology. The impact of alterations in intestinal microbiota and TLR2 signaling on PD pathogenesis in a nontransgenic mouse model of sporadic PD has not yet been elucidated. Furthermore, it is important to note that the effect of TLR2 on the pathogenesis of proteinopathy might differ between models that recapitulate the genetic etiology and those that do not [30,31].

Among nontransgenic mouse models of sporadic PD, the intracerebral injection of α-synuclein preformed fibrils (PFFs) in mice can recapitulate the α-synuclein pathology of PD patients. An administration of α-synuclein PFFs into the striatum can induce the aggregation and phosphorylation of endogenous α-synuclein monomers [8,32,33,34]. Subsequently, the phosphorylated α-synuclein aggregates propagate from the striatum to other regions, leading to microglial activation [8,33,34].

Therefore, in the present study, we aimed to determine the impact of alterations in intestinal microbiota and TLR2 signaling on α-synuclein pathology in nontransgenic mice administered α-synuclein PFFs as a model of PD due to nongenetic causes.

## 2. Materials and Methods

### 2.1. Animals

In the present study, we used male C57BL6/J mice (Charles River Laboratories Japan, Kanagawa, Japan), C57BL6/J Jcl mice (CLEA Japan, Tokyo, Japan) and TLR2-KO mice in a C57BL6/J Jcl background (Oriental Bio Service, Kyoto, Japan). All animals were housed in controlled 12:12 light–dark cycles and had access to diet and water ad libitum. In experiments investigating the effect of antibiotics, all animals had access to sterilized diet and water ad libitum. All animal experiments were performed in accordance with the regulations for animal experiments and related activities of the Institutional Animal Care and Use Committee of Sumitomo Pharma (Approval number: AN12360-B01 and AN12410-B00).

### 2.2. Stereotaxic Injection of α-Synuclein PFF into Striatum

Recombinant human α-synuclein fibrils (Cosmo Bio, Tokyo, Japan) were concentrated to 5 mg/mL using Amicon^®^ Ultra-0.5 centrifugal filter units (Merck Millipore, Tullagreen, Ireland). A slightly modified, previously described protocol [35] was used to inject α-synuclein PFF or phosphate-buffered saline (PBS) as vehicle into the striatum of 7–9-week-old mice. Briefly, mice were anesthetized with isoflurane, and the skin was disinfected using a 10% povidone–iodine solution (Shionogi Pharma, Osaka, Japan). α-synuclein PFF or PBS was injected into the right dorsal striatum (A-P, 0.2 mm; lateral, −2.0 mm; P-V, −2.6 mm from bregma and dura) using a 10 µL Neuros syringe with a 33-gauge needle (Hamilton Company, Reno, NV, USA). Two microliters of α-synuclein PFF or PBS was injected at a rate of 0.2 µL/min for a total of 10 µg/animal. After injection, the needle was kept in place for one more minute for proper diffusion of the solution. The surgical incision was closed using surgical adhesive glue (Aron Alpha A “Sankyo”; Daiichi Sankyo Company, Tokyo, Japan). After the mice recovered from surgery, they were placed back into their home cages and monitored regularly.

### 2.3. Immunohistochemistry

Mice were euthanized under isoflurane anesthesia and perfused with PBS, followed by perfusion with 4% paraformaldehyde (FUJIFILM Wako Pure Chemical). To prepare frozen sections, the brains were removed and immersed in 4% paraformaldehyde at 4 °C overnight, followed by immersion in 20% (*w*/*v*) sucrose solution at 4 °C for 24 h. The fixed brains were embedded into the O.C.T. compound (Sakura Finetek Japan, Tokyo, Japan), and 10 μm thick cryostat sections were prepared. To prepare paraffin-embedded sections, the removed brains were immersed in 10% neutral buffered formalin, followed by dehydration. Fixed brains were embedded in paraffin wax, and 6 μm thick sections were prepared.

Antibodies used for immunohistochemical analyses are listed in Table 1. The paraffin-embedded sections were deparaffinized and rehydrated by incubation with xylene and a series of graded ethanol baths. To retrieve antigen, the sections were boiled in Target Retrieval Solution (pH 6.1; Agilent Technologies Japan, Tokyo, Japan) for 10 min. After cooling down, the sections were rinsed with Tris-buffered saline (TBS) and then incubated with 3% (*v*/*v*) H_2_O_2_ in distilled water to block endogenous peroxidase activity. Following rinsing with distilled water, the sections were blocked with 4% (*w*/*v*) Block Ace solution (KAC, Kyoto, Japan) in distilled water and incubated for 1 h at room temperature. Incubation with appropriate primary antibodies was performed in IMMUNO SHOT immunostaining, Fine (Cosmo Bio). After incubation of the sections with appropriate secondary antibodies, staining was detected using 0.02% (*w*/*v*) 3,3′-diaminodenzidine (DOJINDO LABORATORIES, Kumamoto, Japan) in TBS. The sections were counterstained with hematoxylin (Sakura Finetek Japan) and dried with cold air. After immersing in xylene, the slides were covered with a coverslip using Tissue-Tek Glas (Sakura Finetek Japan). Images of sections were captured using an Aperio ScanScope device (Leica Microsystems, Tokyo, Japan). To assess α-synuclein accumulation and microglial activation, number of phosphorylated α-synuclein-positive aggregates and percentage of Iba1-positive area per regional area examined were measured, respectively. For measuring Iba1-positive area and regional area examined, the ImageScope software (version 12.4.2; Leica Microsystems) was used.

### 2.4. Antibiotic Treatment

As previously described [36], mice were treated with antibiotics, which were administered daily starting from seven days before the injection of α-synuclein PFF until the end of the experiments. The antibiotics were administered as a cocktail provided in sterile drinking water and included ampicillin (1 g/L; Nacalai Tesque, Kyoto, Japan), vancomycin (0.5 g/L; Shionogi Pharma), neomycin (0.5 g/L; Nacalai Tesque), gentamycin sulfate (100 mg/L; Nacalai Tesque) and erythromycin (10 mg/L; Sigma-Aldrich Japan, Tokyo, Japan). We prepared stock solutions of ampicillin, vancomycin and neomycin by dissolving them in sterilized water at a concentration of 50 mg/mL. Erythromycin was dissolved in 99.5% ethanol (FUJIFILM Wako Pure Chemical) at a concentration of 50 mg/mL, while gentamycin was dissolved in sterilized water at concentration of 25 mg/mL. The stock solutions were aliquoted and stored at −30 °C for two months. For preparation of an antibiotic cocktail, the stock solutions were mixed with 300 mL of sterilized water. The average amount of water intake was 5 to 7 mL/mouse/day.

### 2.5. Treatment with a TLR1/2 Agonist

A TLR1/2-specific agonist CU-T12-9 (Tocris Bio-Techne Japan, Tokyo, Japan) dissolved in 0.5% (*w*/*v*) Methyl Cellulose 400 (FUJIFILM Wako Pure Chemical, Osaka, Japan) was used to treat mice. CU-T12-9 or 0.5% methylcellulose (dosing volume, 10 mL/kg) was administered intraperitoneally. The individual dosing volume was calculated based on the body weight measured on the experimental day. Two hours after dosing, mice were euthanized under isoflurane anesthesia and perfused with chilled PBS. Removed brains were used for quantitative polymerase chain reaction.

### 2.6. Counting of Intestinal Bacteria

Fecal samples collected from each mouse were suspended in brain–heart infusion (BHI) medium (Difco Laboratories, Detroit, MI, USA) at a 1:19 (*w*/*v*) ratio. To count intestinal bacteria, 10-fold serial dilutions of fecal suspensions were plated on BHI agar (Difco Laboratories). After 48 h of anaerobic incubation at 37 °C, the number of colony-forming units was counted. For anaerobic incubation, Anaero-Pack (Mitsubishi Gas Chemical, Tokyo, Japan) was used.

### 2.7. Quantitative Polymerase Chain Reaction

Mice were euthanized under isoflurane anesthesia and perfused with chilled PBS. Brains were removed and immersed in a solution containing RNAlater (Thermo Fisher Scientific, Tokyo, Japan) and incubated overnight at 4 °C. Immersed brains were homogenized using Micro Smash (Tomy Seiko, Tokyo, Japan). Total RNA was extracted from tissue homogenates using the RNeasy mini kit (Qiagen, Germantown, MD, USA), and cDNA was synthesized from 1 µg of RNA using the TaqMan Fast Cells-to-CT kit (Thermo Fisher Scientific). The concentration of RNA in each sample was determined using NanoDrop (Thermo Fisher Scientific). The TaqMan real-time polymerase chain reaction assays using probes and primers for *Tlr2*, *Tnf* and 18S rRNA (Applied Biosystems, Waltham, CA, USA) were performed using the QuantStudio 7 Flex system (Applied Biosystems). For an endogenous reference gene, 18S rRNA was used. Fold changes were determined using the ΔΔCT method.

### 2.8. Statistical Analysis

Statistical differences among groups were determined using one-way analysis of variance, followed by Dunnett’s test. Statistical differences between two groups were determined using Student’s *t* test. A *p* value of <0.05 was considered significant.

## 3. Results

### 3.1. Antibiotic Treatment Does Not Have a Significant Effect on α-Synuclein Pathology

In the present study, we used wild-type (WT) mice injected with α-synuclein PFF as a nongenetic model of PD and evaluated α-synuclein accumulation and microglial activation as readouts for α-synuclein pathology. In this model, the striatal administration of α-synuclein PFF led to significant increases in the number of phosphorylated α-synuclein (p-α-synuclein)-positive aggregates and in the Iba1-positive area in the cortex of vehicle-treated mice (Figure 1A,B). In these mice, p-α-synuclein accumulated in neurites and partially in soma (Figure 1C). These results clearly indicated that our model was consistent with those reported in previous studies [8,33,35] and that it was suitable for evaluating the effects of intestinal microbiota on α-synuclein pathology in the brain.

Next, to evaluate the effects of intestinal microbiota on α-synuclein pathology in the brain, we investigated whether a reduction in the density of intestinal microbiota by antibiotic treatment impacted α-synuclein accumulation in the brain. Although antibiotic treatment led to a small reduction in the number of p-α-synuclein-positive aggregates in the brain, the number of p-α-synuclein-positive aggregates and the Iba-1 positive area was not significantly different between the antibiotic-treated and vehicle-treated mice (Figure 1A,B). The density of intestinal microbiota was significantly lower in the antibiotic-treated mice than in the vehicle-treated mice (Figure 1E).

### 3.2. TLR2 Knockout Does Not Have a Significant Impact on α-Synuclein Pathology

Based on the reported role of α-synuclein-positive aggregates as TLR2 ligands, we hypothesized that TLR2 would impact PD pathology independent of the intestinal microbiota. To this end, we injected α-synuclein PFF into TLR2-KO mice and examined the accumulation of α-synuclein aggregates and the Iba1-positive area in comparison to the WT mice. Although TLR2-KO led to a small reduction in the number of p-α-synuclein-positive aggregates and the Iba1-positive area, we found no significant differences between the TLR2-KO and WT mice after the injection of α-synuclein PFF (Figure 2A,B). Treatment with a TLR1/2-specific agonist, which increased *Tlr2* mRNA levels in the brains of WT mice, did not lead to an increase in *Tlr2* mRNA levels in the brains of TLR2-KO mice (Figure 2E), indicating that the TLR2-mediated signaling pathway was indeed suppressed in the TLR2-KO mice.

### 3.3. Tlr2 and Tnf Expression Levels Are Not Robustly Elevated in α-Synuclein PFF-Injected Mice

To elucidate factors that might be associated with the effects of intestinal microbiota and TLR2 signaling on α-synuclein pathology, we evaluated the *Tlr2* and *Tnf* mRNA levels. As shown in Figure 3A,B, the *Tlr2* and *Tnf* mRNA levels were 1.8- and 1.5-fold higher in the α-synuclein PFF-injected mice than in the PBS-injected mice, respectively, with a significant difference observed only in *Tlr2* expression levels.

## 4. Discussion

Several studies have reported the involvement of intestinal microbiota and TLR2 signaling in PD pathology using genetic models of PD. Furthermore, some studies have also reported that the effects of TLR2 signaling on proteinopathy might depend on the presence of genetic factors [30,31]. Although sporadic PD does not necessarily have a genetic etiology, the impact of alterations in intestinal microbiota and TLR2 signaling on PD pathogenesis in the nontransgenic mouse model of sporadic PD has not been extensively investigated. Therefore, in the present study, we aimed to investigate the contribution of intestinal microbiota and TLR2 signaling on PD pathology in a nontransgenic model that utilized α-synuclein PFF injection to further our understanding of sporadic PD pathogenesis by excluding genetic etiology.

Our analyses of nontransgenic mice injected with α-synuclein PFF revealed that in the absence of a genetic etiology, antibiotic treatment and TLR2 knockout led to a small and nonsignificant reduction in the number of p-α-synuclein-positive aggregates and the Iba1-positive area (Figure 1 and Figure 2). These histopathological changes, which we utilized as indicators to evaluate the effect of α-synuclein PFF injection, partially replicated the pathologic changes observed in *SNCA* transgenic mice [37] and in patients with PD [27], indicating the relevance of our model in evaluating PD pathology. The observed effects of intestinal microbiota and TLR2 signaling on α-synuclein pathology in the present study were different than those reported in *SNCA* transgenic models that recapitulate the genetic etiologic factors of PD. We speculate that the observed difference between the present study and other models might be attributed to the involvement of various pathways in pathologic processes related to α-synuclein accumulation. Among these pathways, the presence of genetic etiologic factors of PD might impact the effects of intestinal microbiota and TLR2 signaling on the pathologic processes related to α-synuclein accumulation.

Our hypothesis is supported by similar results reported in models of Alzheimer’s disease, which is also a proteinopathy. The effects of TLR2 signaling on the accumulation of amyloid β (Aβ) aggregates, which also act as TLR2 ligands [38], in the brain differ among models with genetic etiology (amyloid precursor protein/presenilin 1 transgenic model) and those without genetic etiology (nontransgenic model of Aβ fibril injection) [30,31]. This difference might be attributed to alterations in specific processes involved in the accumulation of Aβ aggregates, such as the extracellular secretion of Aβ_1–42_ [39], which might be influenced by the presence of mutations in amyloid precursor protein or presenilin 1. Mutations in *SNCA* also impact the aggregation of α-synuclein. For example, the neuronal axonal transport of α-synuclein [40] and the production of cytokines such as TNFα by microglia [41,42], which influence the aggregation of α-synuclein [43], vary depending on the presence of *SNCA* mutations. In addition, in vivo α-synuclein PFF injection has been demonstrated to induce more severe α-synuclein propagation and accumulation in *SNCA* transgenic mice than in nontransgenic mice [8]. Based on these findings, it is possible that the differences in α-synuclein accumulation due to genetic factors might also impact the effects of intestinal microbiota and TLR2 signaling on α-synuclein pathology.

Various pathways have been reported as mechanisms by which intestinal microbiota and TLR2 signaling affect α-synuclein accumulation in PD [44]. Studies have shown that microglial activation in PD pathology can induce the accumulation of α-synuclein aggregates and the degeneration of dopaminergic neurons in the substantia nigra [45,46]. Changes in the composition of intestinal microbiota have also been shown to alter the production of short-chain fatty acids [36] and TLR2 ligands [24], which in turn induce microglial activation in the central nervous system and the subsequent accumulation of α-synuclein aggregates [36,47]. The accumulation of α-synuclein aggregates in the brain activates TLR2 signaling, which further promotes α-synuclein accumulation [23,48]. Previous reports have shown that a decrease in the density of intestinal microbiota or TLR2 knockout inhibits microglial activation, increases *Tlr2* expression levels and leads to the accumulation of α-synuclein aggregates in the brain [23,36,48]. Therefore, we evaluated microglial activation and *Tlr2* expression levels, which are considered important factors in α-synuclein accumulation. Indeed, we observed significant increases in *Tlr2* expression levels and the Iba1-positive area (Figure 1B, Figure 2B and Figure 3A), indicating that our model successfully recapitulated PD pathology observed in patients [27] and *SNCA* transgenic mice [23,28]. However, the mRNA levels of *Tnf* were not significantly higher in the α-synuclein PFF-injected mice than in the PBS-injected mice, despite a trend of increase (Figure 3B). The degree of increase in *Tnf* mRNA levels was smaller than that observed in *SNCA* transgenic mice, in which approximately six-fold increase was reported in one study [23]. TNFα is an inflammatory cytokine that promotes α-synuclein accumulation [43], and its increase can be inhibited by reducing the density of intestinal microbiota or knocking out *Tlr2* in *SNCA* transgenic mice [23,36]. Therefore, the degree of TNFα increase in a given model might impact the effect of intestinal microbiota and TLR2 signaling on pathologic processes related to α-synuclein accumulation. However, TNFα levels would not necessarily increase in the brains of patients with PD. In fact, several study shows that the concentrations of TNFα in cerebrospinal fluid were reported to be comparable between patients with sporadic PD and healthy, age-matched controls [49]. Overall, our findings indicate that these aspects of sporadic PD can be reproduced in α-synuclein PFF-injected mice used in the present study.

In the present study, we evaluated the impacts of intestinal microbiota and TLR2 signaling on the progression of the pathology related to α-synuclein accumulation triggered by α-synuclein PFF administration and found that their impacts were small in the absence of a genetic etiology. In relation to this issue, it has been shown that functional amyloid proteins (curli) produced by several bacterial species within the intestinal microbiota can promote the formation of α-synuclein aggregates [50]. Therefore, for a better understanding of the impact of intestinal microbiota and TLR2 signaling on PD pathology in the absence of a genetic etiology, further studies to determine their impact on the formation of α-synuclein-related pathology are needed.

## 5. Conclusions

The main result of the present study is that the impact of intestinal microbiota and TLR2 signaling on the pathology related to α-synuclein accumulation was small in the absence of a genetic etiology. These results and previous findings in *SNCA* transgenic mice indicate that the contribution of intestinal microbiota and TLR2 signaling to α-synuclein pathogenesis might be influenced by genetic factors. Considering the diverse pathways leading to the pathogenesis of PD in humans, the model used in the present study is suitable for investigating novel factors that may be involved in the pathogenesis of sporadic PD, which is difficult to reproduce in transgenic models based on genetic etiology. The present study’s findings indicate that the integration of insights obtained in this experimental model with those observed in models of genetic etiology will advance our understanding of the molecular mechanisms underlying sporadic PD pathogenesis.

## Figures and Tables

**Figure 1 microorganisms-12-00106-f001:**
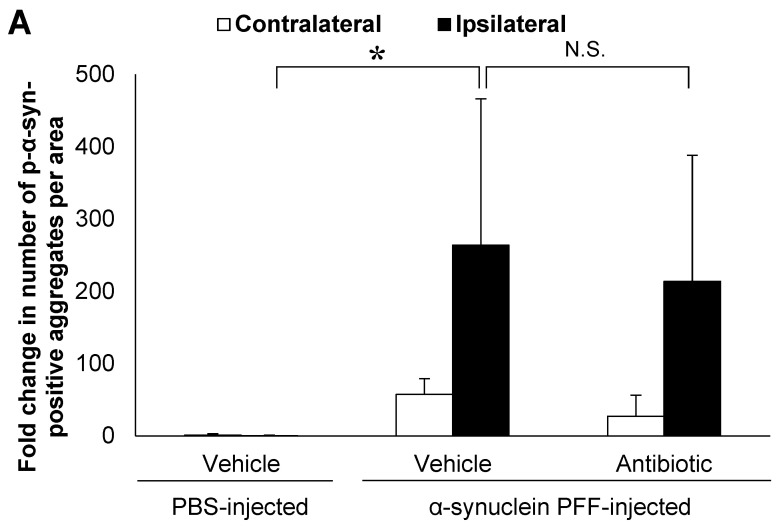
Alpha-synuclein pathology in the cortex of nontransgenic mice injected with α-synuclein preformed fibrils and the effect of antibiotic administration. Mice were euthanized 1.5 months after the unilateral injection of α-synuclein preformed fibrils (PFF) or vehicle into the striatum, and frozen brain sections (10 μm thick) were immunostained with antibodies against α-synuclein phosphorylated at S126 (p-α-syn; (**A**,**C**)) or Iba1 (**B**,**D**). Data are shown as means ± standard error of fold changes in (**A**) the number of p-α-syn-positive aggregates and (**B**) the percentage of Iba1-positive area per cortical area. *n* = 3 mice/group. * *p* < 0.05, ** *p* < 0.01, compared to α-synuclein PFF-injected, vehicle-treated group by Dunnett’s test. N.S., not significant. Representative images for immunohistochemical staining for p-α-syn (**C**) and Iba1 (**D**). Arrows show p-α-syn-positive aggregates or Iba1-positive microglia. Scale bar, 200 μm. Viable counts of bacteria after antibiotic treatment are shown as means ± standard deviation of colony forming units (CFU), determined by anaerobic incubation of fecal samples from mice (**E**). *n* = 3 mice/group. ** *p* < 0.01, compared to antibiotic-treated group at the post-treatment stage by Dunnett’s test.

**Figure 2 microorganisms-12-00106-f002:**
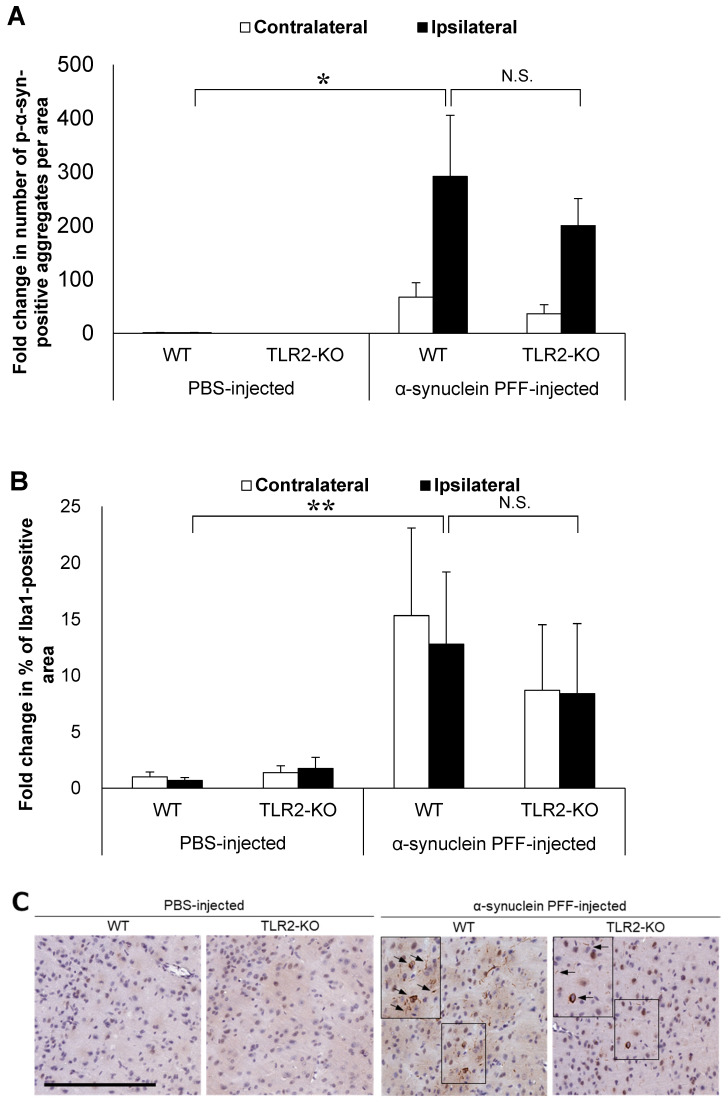
Impact of Toll-like receptor 2 knockout on α-synuclein pathology in the cortex of α-synuclein PFF-injected nontransgenic mice. Mice were euthanized 1.5 months after the unilateral injection of α-synuclein PFF or vehicle into the striatum, and paraffin-embedded brain sections (6 μm thick) were immunostained with antibodies against α-synuclein phosphorylated at S126 (p-α-syn; (**A**)) or Iba1 (**B**). Data are shown as means ± standard error of fold changes in (**A**) the number of p-α-syn-positive aggregates and (**B**) the percentage of Iba1-positive area per cortical area. *n* = 3–5 mice/group. * *p* < 0.05, ** *p* < 0.01, compared to α-synuclein PFF-injected WT mice using Dunnett’s test. N.S., not significant. Representative images for immunohistochemical staining for p-α-syn (**C**) and Iba1 (**D**). Scale bar, 200 μm (**C**) and 100 μm (**D**). Arrows show p-α-syn-positive aggregates or Iba1-positive microglia. Expression levels of *Tlr2* in the cortex of TLR2-KO mice were determined using quantitative real-time polymerase chain reaction (**E**). Mice were euthanized 2 h after administration of TLR1/2 agonist (CU-T12-9) or vehicle. Data are presented as means (bars) and individual values (white circles) of fold changes in the mRNA levels of *Tlr2* in the cortex. *n* = 2 mice/group.

**Figure 3 microorganisms-12-00106-f003:**
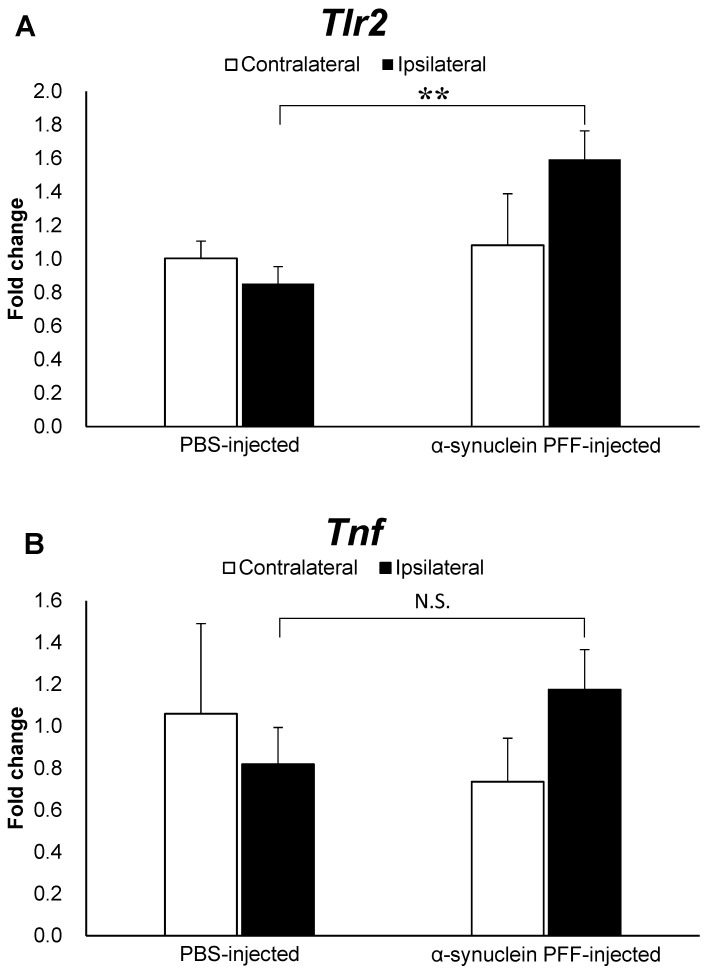
Expression levels of *Tlr2* and *Tnf* in the cortex of α-synuclein PFF-injected nontransgenic mice. Quantitative real-time polymerase chain reaction was used to determine the expression levels of *Tlr2* and *Tnf* in the cortices of mice, which were euthanized 1.5 months after the unilateral injection of α-synuclein PFF or vehicle into the striatum. Data are presented as means ± standard deviation of fold changes in the mRNA levels of *Tlr2* (**A**) and *Tnf* (**B**) in the cortex. *n* = 3 mice/group. ** *p* < 0.01 between the vehicle-injected and α-synuclein PFF-injected groups using Student’s *t* test. N.S. means not significant.

**Table 1 microorganisms-12-00106-t001:** List of antibodies used in the present study.

Antibodies	Supplier	Catalog Number	Dilution
Anti-Alpha Synuclein (phospho S129)	Abcam (Cambridge, UK)	ab51253	1:1000
Anti-Iba1	FUJIFILM Wako Pure Chemical (Osaka, Japan)	019-19741	1:1000
Simple Stain Mouse MAX PO (R)	NICHIREI BIOSCIENCES (Tokyo, Japan)	414341	No dilution

## Data Availability

Data are available from the corresponding author upon request.

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
