# Peer review of "The Impact of Intestinal Microbiota and Toll-like Receptor 2 Signaling on α-Synuclein Pathology in Nontransgenic Mice Injected with α-Synuclein Preformed Fibrils"

_microorganisms, 2024, doi:10.3390/microorganisms12010106_

Round 1

Reviewer 1 Report

Comments and Suggestions for Authors

The present manuscript is a well organized research article. The authors did a great job! 

I would like to ask to add some more details. 

Please add the concentration of the antibodies used. 

Please add some labels in the figures to show the differences between the groups. 

Please add a limitation of the research part. 

Please add a separate conclusion if possible. 

Author Response

Thank you very much for taking your time to review our manuscript. Please see the attachment to confirm our response to your helpful comments.

Reviewer 2 Report

Comments and Suggestions for Authors

In the article by Koyanagi et al., entitled “Impact of intestinal microbiota and toll-like receptor 2 signaling on α-synuclein pathology in nontransgenic mice injected with α-synuclein preformed fibrils” demonstrated that the impact of intestinal microbiota and TLR2 signaling on the pathology related to α-synuclein accumulation was small in the absence of a genetic etiology. These results also indicate that the contribution of intestinal microbiota and TLR2 signaling to α-synuclein pathogenesis might be influenced by genetic factors. This article is interesting to people working in the field of Parkinson’s disease. However, the authors need to address all the queries raised by the reviewers.

Comments:

- Lines 210-211: Figure 1A covers some words from line 210 and should be adjusted downwards appropriately. Please make sure your materials are properly prepared and formatted before submitting revisions.

- Line 214: Immunostaining in the α-synuclein PFF-injected group is less obvious and less abundant than in the PBS-injected group in Figure 1D. Is it correct?

- Line 231: As mentioned in the article, antibiotic-treated mice had significantly lower gut microbiota density than vehicle-treated mice. Can the author add additional image data?

- Lines 239-242: From the article, Tlr2 mRNA levels in the brains of WT mice can be increased by treatment with TLR1/2-specific agonists, but Tlr2 mRNA levels in the brains of TLR2-KO mice do not Increase. Can the authors add additional images from the data?

- Lines 248-249: The annotation for Figure 2 mentions that brain tissue sections have immunostaining, can the authors add any other image data?

The authors should consider the limitations of this study. It can provide a basis for further in-depth research.

Author Response

(The authors gave the same response as above.)

Round 2

Reviewer 2 Report

Comments and Suggestions for Authors

To meet publication requirements, each comment has been completely revised.